# Effect of additional Nordic hamstring exercise or sprint training on the modifiable risk factors of hamstring strain injuries and performance

**Nicholas J. Ripley**[1]*, **Matthew Cuthbert**[1,2], **Paul Comfort**[1,3], **John J. McMahon**[1]

1 Human Performance Laboratory, University of Salford, Salford, United Kingdom, 2 The Football Association Group, St George's Park, Burton-upon-Trent, Staffordshire, United Kingdom, 3 School of Medical and Health Sciences, Edith Cowan University, Joondalup, Australia

These authors contributed equally to this work.
* n.j.ripley@salford.ac.uk

**Data Availability Statement:** All relevant data are within the manuscript and its Supporting Information files.

## Abstract

The Nordic hamstring exercise (NHE) has commonly been investigated in isolation, however, within practice multiple modalities are commonly incorporated. However, the NHE has a low level of compliance within sport, with sprinting being potentially being preferred. The present study aimed to observe the effect of a lower-limb program with either additional NHE or sprinting on the modifiable risk factors hamstring strain injury (HSI) and athletic performance. 38 collegiate athletes were randomly assigned into three groups: control standardised lower-limb training program ($n$ = 10 (2 female, 8 male), age = 23.50 ± 2.95 years, height = 1.75 ± 0.09 m, mass 77.66 ± 11.82 kg), additional NHE ($n$ = 15 (7 female, 8 male), age = 21.40 ± 2.64 years, height = 1.74 ± 0.04 m, mass 76.95 ± 14.20 kg) and additional sprinting ($n$ = 13 (4 female, 9 male), age = 22.15 ± 2.54 years, height = 1.74 ± 0.05 m, mass 70.55 ± 7.84 kg). All participants performed a standardised lower-limb training program twice per week for seven weeks, including Olympic lifting derivatives, squatting movements, and the Romanian deadlift, with experimental groups performing with either additional sprinting or NHE. Bicep femoris architecture, eccentric hamstring strength, jump performance, lower-limb maximal strength and sprint ability were measured pre and post. All training groups demonstrated significant ($p$ < 0.001), small-moderate increases in Bicep femoris architecture ($g$ = 0.60–1.22), with significant ($p$ < 0.001), small-large increases in absolute and relative eccentric peak force ($g$ = 0.60–1.84). Significant and small increases were observed in take-off velocity and mean propulsion force ($p$ < 0.02, $g$ = 0.47–0.64), with non-significant and small increases for both the sprint and control training groups for mean propulsion force ($p$ > 0.05, $g$ = 0.42–0.50). Nordic and sprint training groups had significant and small increases in peak absolute and relative net force ($p$ < 0.001, $g$ = 0.44–0.60). The control group had a non-significant trivial increase in absolute peak net force ($p$ > 0.05, $g$ = 0.22), with a significant and small increase in relative peak relative net force ($p$ = 0.034, $g$ = 0.48). Significant and small decreases for the NHE and sprinting training groups was observed for 0–10 m, 0–20 m, and 10–20 m sprint time ($p$ < 0.010, $g$ = 0.47–0.71). Performing multiple

**Funding:** This research was funded by the National Strength and Conditioning Association (NSCA) Foundation Doctoral research grant. NJR is the recipient. The funders had no role in study design, data collection and analysis, decision to publish, or preparation of the manuscript.

**Competing interests:** The authors have declared that no competing interests exist.

modalities, with either additional NHE or sprinting, as part of a complete resistance training program was superiorly effective for measures of modifiable risk factors HSI, with similar increases observed in measures of athletic performance derived from the standardised lower-limb training program.

## Introduction

Across the literature, training interventions that have attempted to reduce hamstring strain injury (HSI) incidence, have aimed to mitigate the influence of the modifiable risk factors of HSI (i.e., eccentric hamstring strength and bicep femoris long head ($BF_{LH}$)fascicle length (FL)), by targeted exercises, such as the nordic hamstring exercise (NHE) [1–4] or as a combination of exercises (i.e. FIFA 11/11+ warm up protocol [5]). Incorporating the NHE has a meaningful ability to decrease the occurrence of HSI, however, the effectiveness of any intervention modality relies upon the compliance of the athletic population [6, 7], with ≥75% compliance showing superior effectiveness within the literature [7]. Low levels of compliance within studies that have utilised the NHE as part of training interventions have frequently been reported due delayed onset muscle soreness (DOMS) and/or poor athlete support. This is despite only a moderate level of DOMS being reported within NHE training interventions [1, 8]. Furthermore, the NHE; one of the most extensively researched eccentric hamstring exercises, is continually poorly adopted within elite European soccer [9], despite showing superior effectiveness [4, 7, 10]. Bahr, Thorborg and Ekstrand [9], cited high levels of both player and coach complaints when implementing the NHE. One possible explanation is that many players and coaches do not fully understand the potential benefits of implementing the NHE, with many unconvinced of key intervention outcomes (i.e., the NHE reduces injuries, increases player availability, return to play sooner post-HSI) [9].

Currently, the NHE has been a key focus of training research by observing its effect on one or more of the modifiable risk factors of HSI (i.e., eccentric strength, muscle architecture) [1, 6, 11]. Interventions that have utilised the NHE have shown large and significant positive adaptations in both eccentric strength capabilities (isokinetic and Norbord) and $BF_{LH}$ muscle architecture (i.e., increased $BF_{LH}$ FL and decreased pennation angle) [8]. A recent systematic review and meta-analysis, highlighted that the application of the NHE has generally coincided with extremely high volumes, with many interventions progressing to ≥100 repetitions per week—prescribing sets of between 8–12 repetitions [8]. This is despite the NHE being classified a 'supra-maximal' eccentric exercise, of a greater intensity than an equivalent concentric action. Furthermore, as the aim of including the NHE should be to increase the force generating potential of the hamstrings (i.e. increase strength), the current prescription would not fall within the repetition and volume guidelines for the implementation of strength training [12]. More recent research has adopted a low volume approach to NHE training (2 x 4 repetitions performed twice per week [13]), increasing eccentric hamstring strength and $BF_{LH}$ FL, to a similar magnitude as higher volume equivalents, while being more aligned with volume recommendations for strength training. Although similar volumes of training had small to trivial effects upon eccentric isokinetic hamstring strength and $BF_{LH}$ muscle architecture in elite youth soccer players [14]. However, contrastingly in elite senior soccer players 1 set of 3 repetitions had a meaningful effect on eccentric hamstring strength [15], with compliance (or more specifically frequency of stimulus) having a significant role in improvements in eccentric hamstring strength [15].

As a result of the continued low compliance of NHE training, a natural progression of practice and research is to investigate the possibility of training that could be more agreeable or available for both athletes and coaches. One example could be sprint training, as it has been hypothesised there could be a similar imposed demand of fascicle lengthening (i.e. eccentric muscle action), while coinciding with the maximal activation patterns during the swing phase [16–21], which is potentially indicative of the desired adaptive response (i.e. increased eccentric strength and $BF_{LH}$ FL). Furthermore, maximal sprinting has the potential to strengthen the elastic properties of connective tissue, increase motor unit activation, increase passive tension of the muscle-tendon complex and improve cross bridge mechanics, which are all associated with the occurrence of injuries and overall athletic performance [22].

To date, two studies have observed the effects of a sprint-based training on the modifiable risk factors for HSI [23, 24]. Freeman and colleagues [23] observed a positive adaptive response in eccentric hamstring strength from sprint training. Both sprint and NHE training provided a small but significant, positive response to eccentric hamstring strength–although on closer inspection, the NHE training group, who started stronger, displayed a greater adaptive response than the weaker sprint group (9.8- Vs 6.2%Δ) [23]. This indicates that although both groups improved, the NHE was superior [23], it should also be noted that this study was performed across a short duration of four-weeks, where there was no control of other resistance training–both of which could influence the observed response. More recently, Mendiguchia, Conceicão [24] performed a similar study by observing the effect of either the NHE or sprint training upon $BF_{LH}$ architecture. Interestingly, the sprint training group had a moderate, positive increase in $BF_{LH}$ FL, whereas the NHE training only resulted in a small, positive increase in $BF_{LH}$ FL [24], with a 16.21- vs. a 7.38% change, respectively. Although methodological aspects to explain these findings, firstly, the NHE training could be described as being sub-optimal, as there was no progression of eccentric intensity, following a previously established protocol (first six weeks of the study by Petersen et al. [25]). Secondly, the sprint training intervention was quite intensive with multiple sessions of high volumes, even in comparison to the earlier study by Freeman [23], although it would likely be impossible to equate volumes between modalities with a number of complex variables that would need to be considered (including, muscle action type, muscle action time under tension, stride length, stride frequency, repetitions and distances).

Improvements in athletic performance (e.g., strength, sprinting and jumping) are also a key if not the primary consideration when programming for athletes. It is well documented that sprint-based training can improve athletic tasks [22, 26, 27]. Likewise, improvements in both sprint, jump and change of direction performance have also been observed following a NHE intervention [14, 28–31], although the research is inconclusive regarding athletic performance improvements [14, 32]. It has been hypothesized that increases in athletic performance, as a result of NHE interventions, are the result of an increased force generating capacity during hip extension [33], although it is not a well-established theory [32]. Therefore, both sprint and NHE training modalities have the potential to increase performance in athletic tasks, as well as mitigating the risk of HSIs via the improvement of the modifiable risk factors. However, some researchers continually neglect the fact that the aim of the NHE is to mitigate the risk of HSIs, via improvement in both eccentric hamstring strength and $BF_{LH}$ FL and if the goal is to target improvements in hip extension force generating capacity a specific exercise such as the Romanian deadlift could be considered ideal. Thus, conducting a randomized, parallel training study where additional sprint or NHE training is implemented to a standardised lower limb training program, with measures of hamstring strength, architecture, and performance in dynamic tasks (i.e., sprint, strength and jump performance) taken before and after, would be insightful for practitioners with respect to identifying potential best practice and how multiple elements could compliment a complete training programme.

The purpose of the present study was to determine the effect of a short-term (seven-week) intervention with supplemental sprint or NHE, imbedded within an ecologically valid training programme (group 1. Control training (CT) vs group 2. CT plus NHE vs group 3. CT plus sprinting), on the magnitude of adaptations to the modifiable risk factors, i.e., $BF_{LH}$ muscle architecture and eccentric hamstring strength. In addition, a further aim was to observe the effect of the training intervention on the nature of adaptations to overall athletic performance (sprint, CMJ and lower body strength). It was hypothesised that using a multi-modal approach, with the additional NHE or sprint training, would provide the greatest adaptive response to both modifiable risk factors of HSI ($BF_{LH}$ muscle architecture and eccentric hamstring strength), postulating the greatest adaptive response attained from the CT plus NHE group. In addition, it was hypothesised that for CMJ performance the NHE training group would improve upon the countermovement phase, due to an increase in eccentric hamstring capabilities, whilst all groups would improve both absolute CMJ measures (e.g., jump height, take-off velocity), in addition to measures made during the propulsive phase (e.g., propulsion force and impulse). It is further hypothesised that for sprint-based measures, the sprint training group would have the greatest adaptations in performance in comparison to other training groups. Finally, it is hypothesised that there would be no difference in lower body strength, as all groups would be following the same control resistance training programme (not including the NHE).

## Materials and methods

An intervention study design was employed for the present study (Fig 1), pre-intervention testing was completed for all participants, with a sub-group ($n = 24$) returning on a second occasion to determine between-session reliability. All participants were initially randomly allocated into training groups and then completed a comparable 7-week period of resistance training. Group 1 performed the resistance training as a control, while the remaining groups

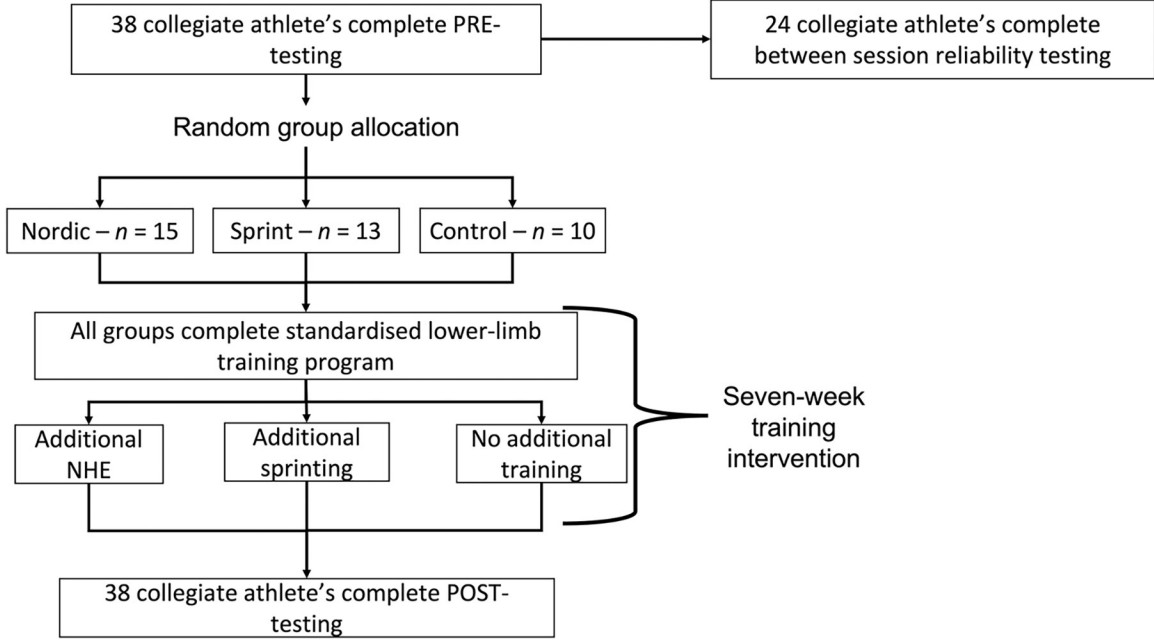

**Fig 1. Schematic diagram of pre-testing, seven-week intervention and post-testing.**

(group 2 & 3) performed an identical resistance training programme with the addition of sprint or the NHE.

## Participants

38 collegiate athletes who participated in regular team sports (football, futsal, rugby union, rugby league, ice hockey, American football, basketball, netball). All participants reported competing across a range of competitive levels from university (collegiate) to semi-professional level sports participation. Participants playing season varied between either pre- or in-season. All participants were required to have a history of resistance-based training, including the NHE, regularly (minimum of once/week) applied within the previous 6 months. All participants reported having between 1–2 years of sprint or running based technical coaching which had been delivered during sport-based training. All participants were required to be free from injury and not had a previous HSI in the past 6 months. Participants were randomly allocated to the three training groups using a random number generator; Nordic $n$ = 15 (7 female, 8 male), age = 21.40 ± 2.64 years, height = 1.74 ± 0.04 m, mass 76.95 ± 14.20 kg, Sprinting $n$ = 13 (4 female, 9 male), age = 22.15 ± 2.54 years, height = 1.74 ± 0.05 m, mass 70.55 ± 7.84 kg, Control $n$ = 10 (2 female, 8 male), age = 23.50 ± 2.95 years, height = 1.75 ± 0.09 m, mass 77.66 ± 11.82 kg. The study was approved by the institutional ethics committee (University of Salford, HSR1819-103). The study also conformed to the principles of the Declaration of Helsinki (1983). None of the participants sustained an injury during the intervention period.

## Procedures

**Training programme.** Participants in the control group and both the intervention groups completed an identical lower limb resistance training programme, performed twice per week. Each resistance training session consisted of three lower limb exercises, where the training volume remained constant across the training intervention, whilst intensity was manipulated (Table 1). This is consistent with a previous intervention observing changes in athletic performance with the addition of a unilateral exercise within the present study [34]. The loads for all exercises were based of self-identified recently achieved training maxes, the power clean and mid-thigh pulls were based of the subjects one repetition max (1RM) power clean, and the

**Table 1. Lower limb resistance training programe, including sets x reps and estimated one repetition maximum percentages, performed by the control and intervention groups across the seven-week training intervention.**

| Day 1 | | | | | | | |
|---|---|---|---|---|---|---|---|
| Weeks | 1 | 2 | 3 | 4 | 5 | 6 | 7 |
| Power clean | 3 x 3 | 3 x 3 | 3 x 3 | 3 x 3 | 3 x 3 | 3 x 3 | 3 x 3 |
| | 80% | 85% | 90% | 75% | 80% | 85% | 90% |
| Back Squat | 3 x 3 | 3 x 3 | 3 x 3 | 3 x 3 | 3 x 3 | 3 x 3 | 3 x 3 |
| | 80% | 82.50% | 85% | 75% | 80% | 82.50% | 85% |
| Reverse lunge | 3 x 6 | 3 x 6 | 3 x 6 | 3 x 6 | 3 x 6 | 3 x 6 | 3 x 6 |
| | 70% | 72.5% | 75% | 70% | 72.5% | 75% | 77.5% |
| Day 2 | | | | | | | |
| Mid-thigh pulls | 3 x 3 | 3 x 3 | 3 x 3 | 3 x 3 | 3 x 3 | 3 x 3 | 3 x 3 |
| | 80% | 85% | 90% | 75% | 80% | 85% | 90% |
| Romanian deadlift | 3 x 6 | 3 x 6 | 3 x 6 | 3 x 6 | 3 x 6 | 3 x 6 | 3 x 6 |
| | 70% | 72.5% | 75% | 70% | 72.5% | 75% | 77.5% |
| Reverse lunge | 3 x 6 | 3 x 6 | 3 x 6 | 3 x 6 | 3 x 6 | 3 x 6 | 3 x 6 |
| | 70% | 72.5% | 75% | 70% | 72.5% | 75% | 77.5% |

**Table 2. Additional training performed by the NHE or sprint intervention groups across the seven-week training intervention, including sets x reps.**

| | Day 1 & 2 | | | | | | |
|---|---|---|---|---|---|---|---|
| **Weeks** | **1** | **2** | **3** | **4** | **5** | **6** | **7** |
| **Nordic** | 2 x 4 | 2x 4 | 2 x 4 | 2 x 4 | 2 x 4 | 2 x 4 | 2 x 4 |
| **Sprint** | 4 x 25 m | 5 x 25 m | 6 x 25 m | 7 x 25 m | 7 x 25 m | 7 x 25 m | 7 x 25 m |

loads prescribed for the remaining exercises were based one predicted 1RM loads based of 3 or 5RM performances in previous phases of training. Immediately post-training, using a numeric scale of 1–10, a rating of perceived exertion (RPE) was obtained from all participants. Approximately 24-hours post-training, using a numeric pain scale of 1–10, a score for DOMS was attained for all participants.

In conjunction to the control resistance training programme, the intervention groups were prescribed either additional sprint or NHE training at the start or end of each training session (Table 2), respectively. The NHE volume was maintained across the seven-week intervention, in accordance with the low volume recommendations by Presland et al. [13]. Where participants were observed to have sufficient strength to completely control the movement in the final 10–20˚ of knee extension during the NHE, they were then required to hold a weight plate to ensure supramaximal exercise intensity was maintained (2.5 kg increments) [1, 11]. The sprint training group initially experienced incremental increases in sprint volume for the first four-weeks, to minimise a large spike in training load to reduce any risk of HSI incidence [35], following the fourth week sprinting volume was maintained. Sprint training was split across the week, where one training day commenced from a static three-point stance whereas on the second training day participants utilised a rolling start, aiming to accelerate into the sprint similar to the prescription by Freeman et al. [23]. A certified strength and conditioning coach was present at all training sessions, providing verbal feedback on the participants' performance and technique.

The study aimed to control for any other resistance training performed by the participants, advising that outside the prescribed programme no further lower-limb resistance training could be performed. Only an individual's sport-specific and upper body resistance training was permitted.

## Data collection

**Bicep femoris long head muscle architecture.** All testing commenced with resting US imaging of the $BF_{LH}$. For the collection of $BF_{LH}$ muscle architecture, initially the scanning site for all images was determined as the halfway point between the ischial tuberosity and the knee joint fold, along the line of the BF. Images were recorded while participants lay relaxed in a prone position, with the hip in neutral and the knee fully extended. Images were subsequently collected along the longitudinal axis of the muscle belly utilizing a 2D, B-mode ultrasound (MyLab 70 xVision, Esaote, Genoa, Italy) with a 7.5 MHz, 10 cm linear array probe with a depth resolution of 67 mm.

To collect the ultrasound images, a layer of conductive gel was placed across the linear array probe; the probe was then placed on the skin over the scanning site and aligned longitudinally to the BF and perpendicular to the skin. During collection of the ultrasound images, care was taken to ensure minimal pressure was applied to the skin, as a larger application of pressure distort images leading to temporarily elongated muscle fascicles. The assessor manipulated the orientation of the probe slightly if the superficial and intermediate aponeuroses were not parallel. These methods are consistent to those used previously [36].

**Countermovement jump.** Following muscle architecture assessment, participants performed a standardised dynamic warm-up consisting of body weight squats, forward and reverse lunges, submaximal squat jumps and CMJs. Three maximal effort CMJs, with a one-minute rest between trials was assessed using a Kistler force platform, sampling at 1000 Hz, with data collected via Bioware 5.11 software (type 9286AA, Kistler Instruments Inc. Amherst, NY, USA). Participants were instructed to stand still for the initial one second of data collection [37, 38] to enable the subsequent determination of body weight (vertical force averaged over one second). Raw unfiltered, force-time data was exported for subsequent analysis. For the CMJ, participants were instructed to perform the jumps as fast and as high as possible, whilst keeping their arms akimbo. Any jumps that were inadvertently performed with the inclusion of arm swing or leg tucking during the flight phase were omitted and additional jumps were performed after one minute of rest.

**Eccentric hamstring strength.** The assessment of eccentric knee flexor strength was performed using the Nordbord device (Vald Performance, Newstead, Australia), which has been used in the literature previously [1, 23, 39–43]. Within the present study, participants knelt upon a padded board, with ankles secured superior to the lateral malleolus by two individual ankle braces. Attached to the ankle braces were uniaxial load cells (50 Hz), allowing for the force generated by the knee flexors during the NHE to be measured. Participants were instructed to perform one set of three maximal NHE repetitions. The instructions to participants were to gradually lean forward at the slowest possible speed while maximally resisting the movement with both limbs, keeping the trunk and hips in a neutral position with the hands held across the chest. Strong verbal encouragement was provided for each subject to provide a maximal effort. An acceptable trial required the force output to reach a distinct peak (indicative of maximal eccentric strength), followed by a rapid decline in force, when the participant was no longer able to resist the gravitational forces [1, 23, 39–43].

**Lower limb maximal strength.** For the isometric mid-thigh pull (IMTP), the procedures and guidelines previously described were used [44]. Each subject adopted a posture that they would use for the start of the second pull phase of the clean, resulting in knee and hip angles of 139.2 ± 2.8˚ and 149.9 ± 3.2˚, respectively. All participants were familiar with this position, through previous performance of weightlifting exercises within training. Joint angles were measured using hand-held goniometer and recorded for standardization. A steel bar which was identical to an Olympic lifting bar, was in a fixed position above the force platform (type 9286AA, Kistler Instruments Inc. Amherst, NY, USA), at a height which replicated the start of the second pull phase of the clean. Participants stood on the force platform with their hands fixed to the bar with lifting straps [44]. Two warm up trials were performed with one-minute rest provided, at 50% and 75% of the participants perceived maximum effort. Once participants had adopted an appropriate position, a countdown of "3,2,1, Pull!" was provided. Minimal pretension (<50 N) was permitted, to ensure minimal slack, prior to initiation of the pull, participants were instructed to pull against the bar, as hard and as fast as possible, pushing their feet into the ground [44]. Two maximal effort trials were performed for approximately five seconds, with strong verbal encouragement provided. Between trials, peak force was required to be within 250 N of each other.

**Sprinting.** Prior to completing the sprint assessment, two 20 m practice sprints at 50- and 75% of perceived maximum intensity, which also served as a brief familiarisation period. Three maximum effort trials of the 20 m sprint were performed, with brief rest periods of two minutes prescribed between trials. Instructions were provided to participants to initiate the sprint from a stationary two-point, split start and to perform a maximal effort throughout the full 20 m [45]. Any sprint trials that were initiated with a countermovement were discarded and supplementary sprint trials were recorded. Brower single-photocell electronic timing gates

(Draper, Utah, USA) were placed at 0 m, 10 m and 20 m increments along an indoor running track, with each emitter and reflector spaced 2 m apart at approximately hip height [45]. Although the initial pair of timing gates were placed at 0 m, the participants started 0.3 m behind this point [45]. Sprint times for each distance were recorded via a handheld computer and the successful maximal effort sprint trials for each participant were taken forward.

## Data analysis

**Bicep femoris architectural digitization.** All sonograms were analysed off-line with Image J version 1.52 software (National Institute of Health, Bethesda, MD, USA). Images were first calibrated to the known field of view (10-cm), then for each image a fascicle of interest was identified. Finally, muscle thickness, pennation angle, observed FL and distance between fascicle end-point and super-fascial aponeurosis were measured three times within each image, to enable complete FL estimation using a previously established reliable linear equation [36].

$$FL = L + (h \div sin(\beta))$$ [1]

Where L is the observable fascicle length, h is the perpendicular distance between the superficial aponeurosis and the fascicles visible end point and $\beta$ is the angle between the fascicle and the superficial aponeurosis.

**Force-time analysis.** Raw force-time data for the CMJ, IMTP and NHE was analysed in Microsoft Excel (Excel 2016, Microsoft, Washington, USA). For the CMJ, velocity of centre of mass at take-off was determined as a measure of performance (take-off velocity) [46], take-off velocity was used in place of jump height, as its measurement error is typically lower. Take-off velocity was determined by dividing vertical force data (minus body weight) by body mass and then integrating the product using the trapezoid rule. The onset of movement for each CMJ trial was considered to have occurred 30 milliseconds prior to the instant when vertical force had decreased by five times the SD of body weight, as derived during the one second silent period [37, 38, 47]. CMJ take-off was identified when vertical force decreased below five times the standard deviation of the force during the flight phase (residual force) [37, 38, 47]. The CMJ phases were identified using the previously established methods [37, 38, 47]. Briefly, the unweighting phase of the CMJ was considered to have occurred between the onset of movement and the instant of peak negative centre of mass velocity. The braking phase of the CMJ was defined as occurring between the instant of peak negative centre of mass velocity and zero centre of mass velocity. The propulsion phase of the CMJ was deemed to have occurred between the instant centre of mass velocity exceeded 0.01 m·s⁻¹ and the instant of take-off. Braking peak force was defined as the maximum value attained during the braking phase. Propulsion mean force was determined as the mean force during the propulsion phase, while impulse was calculated as the area under the net force-time curve (minus body weight) for the propulsion phase using the trapezoid rule [37, 38, 47]. Countermovement displacement and time, was calculated by the combined time or displacement of centre of mass from the initial standing quiet period to the instant of zero centre of mass velocity, achieved at the end of the braking phase. Therefore, including the combined time and displacement of centre of mass during the unweighting and braking phases.

For the IMTP, peak absolute and relative net force was determined as the maximum forces recorded from the whole force-time curve during the IMTP trials [44].

For the NHE, consistent with the IMTP, peak force was determined as the maximum forces recorded from the whole force-time curve. Movement onset was determined as the point when the force increased above a 5 N absolute threshold, whereas the movement was finished

when the vertical force decreased below a 5 N absolute threshold. Total and active impulse were determined by integrating the whole force-time curve and the active portion of the force-time curve (movement onset-finish), respectively. Mean force was determined as the average force across the active portion of the force-time curve. Time to peak force was determined as the time between movement onset and peak force, while repetition time was determined as the time between movement onset and movement finish.

The mean performance of the trials for each assessment was used for further analysis.

## Statistical analyses

Based on investigating changes in both $BF_{LH}$ architecture and eccentric hamstring strength, G*Power (version 3.1.9.2) was used *a-priori* to calculate sample size, please observe the power and sample size statistics below [48]. An effect size of 1.2 was utilised as this magnitude of change used within previous literature [49].

Minimum acceptable Power– 0.80

α– 0.05

*a-priori* sample size– 12 per group

**Reliability and measurement error.** A subsample performed two PRE-testing sessions (n = 24), to determine the between-session reliability and measurement of each variable of interest. All data was first tested using the Shapiro-Wilk test to check if it satisfied parametric assumptions. A two-way random-effects model intraclass correlation coefficient (ICC) and coefficient of variation (CV) with corresponding 95% CI, was used to determine the relative and absolute, respectively. The ICC values were interpreted based on the upper and lower bound CI as ($<0.50$) poor, ($0.5–0.74$) moderate, ($0.75–0.90$) good and ($>0.90$) excellent [50]. Minimum acceptable absolute reliability was confirmed using a CV $<10\%$ [51]. As parametric assumptions were met, a repeated measure analysis of variance (RMANOVA), with post-hoc pairwise comparisons with Bonferroni correction were performed to determine if there was a learning effects between trials (within each session) and between testing sessions (within each week of testing).

The standard error of measurement (SEM) and smallest detectable difference (SDD) for each variable were calculated to establish measurement error scores. The SEM was calculated using the following Formula [2], where $SD_{pooled}$ represents the pooled SD across the two testing sessions:

$$SD_{pooled} \times \sqrt{1 - ICC} \qquad [2]$$

The SDD was calculated using the following established Formula [3]:

$$(1.96 \times \sqrt{2}) \times SEM \qquad [3]$$

As test-retest reliability and measurement error was established for all variables of interest, any observed changes in performance that exceed the associated measurement error would likely be 'true' changes.

**Pre to post intervention changes.** Data obtained at pre was taken forward to perform comparisons at post training, as parametric assumptions were met for all measures using the Shapiro-Wilk test, between- pre and post in the modifiable risk factors ($BF_{LH}$ FL and eccentric hamstring strength), CMJ and IMTP measures were determined via a RMANOVA with post-hoc pairwise comparisons with Bonferroni correction applied. Hedge's *g* ES was calculated to provide a measure of the magnitude of the differences in each variable between trials, sessions and groups and interpreted in line with previous recommendations which defined values

of < 0.35, 0.35–0.80, 0.80–1.5 and > 1.5 as trivial, small, moderate, and large, respectively [52]. Unfortunately, due to unforeseen circumstances, sprint testing was not able to be performed upon the control group, so comparisons are between both experimental groups using the same statistics as above.

All statistical analyses performed using SPSS software (version 25; SPSS Inc. Chicago, IL, USA) with the alpha level set at $P \leq 0.05$. All other statistical analyses will be conducted in Microsoft Excel and Estimationstats.com to create Gardner-Altman estimation plots [53].

## Results

### Reliability and measurement error

Between session reliability and measurement error for all variables of interest are presented in Table 3. All measures achieved acceptable variability, with good-excellent relative reliability was observed for all measures apart from countermovement displacement which showed poor relative reliability.

### Pre- to Post-intervention changes

At pre-testing, there were trivial non-significant differences observed between all groups for bicep femoris fascicle length and eccentric hamstring strength measures (Hedge's $g = 0.03–0.30$, $p > 0.05$). Similarly for measures of athletic performance, there were trivial to small non-significant differences observed at pre-training between groups (Hedge's $g = 0.03–0.541$, $p > 0.05$), with the largest difference being observed in the IMTP where the control group was meaningfully stronger than both experimental groups at pre-intervention.

**Body mass.** Trivial increases ($g < 0.34$) in body mass were observed across all groups from PRE to POST (Table 4).

**Bicep femoris fascicle length.** A non-significant time×training interaction was observed for absolute and relative $BF_{LH}$ FL ($p = 0.236$). Pairwise comparisons revealed significant ($p < 0.001$) and moderate increases in absolute $BF_{LH}$ FL for all training groups (Table 5, Fig 2). The Nordic and sprint training groups displayed the moderate increases in relative $BF_{LH}$ FL, but a small change for the control group (Table 5, Fig 3).

**Eccentric hamstring strength.** Peak and relative peak force demonstrated a significant time×training interaction ($p < 0.01$). Pairwise comparisons revealed significant ($p < 0.001$) and small-large increases in absolute and relative peak force for all training groups (Table 6, Figs 4 and 5).

**Countermovement jump.** A non-significant time×training interaction was observed for take-off velocity and jump momentum (p = 0.834 & 0.518, respectively). Pairwise comparisons, revealed small increases (Table 7, Figs 6 & 7). There were similar percentage increases for take-off velocity (4.44–5.15%) and jump momentum (7.41–9.86) between training groups, interestingly, the control group had the greatest percentage and magnitude of increase across training groups.

All other CMJ variables; countermovement time, displacement, and peak braking force, showed non-significant time×training interaction, with trivial differences from PRE to POST for all training groups.

**Isometric mid-thigh pull.** For peak absolute and relative net force attained from the IMTP assessment, a significant time×training interaction was observed ($p = 0.013$, $p = 0.030$). Pairwise comparisons revealed that the Nordic and sprint training groups had significant and small increases in both peak absolute and relative net force (Table 8, Figs 8 and 9). The control group had a non-significant, trivial increase in absolute peak net force, with a significant and small increase in relative peak relative net force.

**Table 3. Descriptive and reliability statistics for BF$_{LH}$ FL for both 10 cm FOV.**

| | | Mean | SD | CV% | ICC (95% CI) | SEM | SEM% | SDD | SDD% |
|---|---|---|---|---|---|---|---|---|---|
| BF$_{LH}$ FL | 10-cm FOV–Absolute FL (cm) | 9.80 | 0.16 | 1.65 | 0.980 (0.938–0.995) | 0.17 | 1.73 | 0.47 | 4.80 |
| | 10-cm FOV–Relative FL | 0.22 | 0.01 | 3.22 | 0.975 (0.929–0.989) | 0.00 | 1.42 | 0.01 | 3.94 |
| EHS | Peak Force (N) | 326.71 | 16.18 | 4.95 | 0.953 (0.886–0.981) | 12.51 | 3.83 | 34.67 | 10.61 |
| | Relative peak Force (N/kg) | 4.23 | 0.08 | 1.89 | 0.932 (0.877–0.987) | 0.19 | 4.61 | 0.54 | 12.77 |
| CMJ | Countermovement Time (s) | 0.44 | 0.01 | 3.31 | 0.926 (0.823–0.970) | 0.02 | 4.03 | 0.05 | 11.17 |
| | Peak Braking Force (N) | 1962.74 | 27.20 | 1.39 | 0.912 (0.792–0.964) | 109.82 | 5.60 | 304.41 | 15.51 |
| | Countermovement Displacement (cm) | 0.23 | 0.02 | 8.93 | 0.644 (0.293–0.842) | 0.09 | 36.82 | 0.24 | 102.07 |
| | Mean propulsion Force (N) | 1589.25 | 12.64 | 0.80 | 0.981 (0.952–0.992) | 39.42 | 2.48 | 109.28 | 6.88 |
| | Mean propulsion impulse (Ns) | 192.74 | 2.15 | 1.12 | 0.991 (0.976–0.996) | 3.07 | 1.59 | 8.50 | 4.41 |
| | Take off velocity (m/s) | 2.51 | 0.03 | 1.23 | 0.973 (0.933–0.989) | 0.04 | 1.56 | 0.11 | 4.34 |
| IMTP | Peak Net Force (N) | 1743.15 | 6.46 | 0.37 | 0.976 (0.932–0.991) | 127.64 | 7.32 | 212.28 | 12.18 |
| | Peak Relative Net Force (N/kg) | 24.40 | 0.07 | 0.27 | 0.966 (0.928–0.990) | 2.25 | 9.20 | 3.73 | 15.30 |
| SPT | 0–10 m (s) | 1.97 | 0.01 | 0.34 | 0.959 (0.899–0.983) | 0.02 | 0.97 | 0.05 | 2.70 |
| | 0–20 m (s) | 3.22 | 0.01 | 0.20 | 0.980 (0.949–0.992) | 0.02 | 0.62 | 0.06 | 1.72 |
| | 10–20 m (s) | 1.26 | 0.01 | 0.71 | 0.897 (0.759–0.958) | 0.02 | 1.55 | 0.06 | 4.28 |

SD = Standard deviation, CV% = coefficient of variation percentage, ICC = intra-class correlation coefficient, CI = Confidence intervals, SEM = standard error of the measurement, SDD = smallest detectable difference, BF$_{LH}$ FL = bicep femoris fascicle length, EHS = Eccentric hamstring strength, CMJ = countermovement jump, IMTP = isometric mid-thigh pull, SPT = Sprinting

**Sprinting.** Non-significant time×training interactions were observed for sprint and Nordic training groups for 0-10-, 0-20- and 10–20 m ($p = 0.980$, $p = 0.699$, $p = 0.282$, respectively). Pairwise comparisons revealed significant and small decreases for both training groups for 0–10 m, 0–20 m, and 10–20 m sprint time (Table 9, Figs 10–12).

No significant group×time interactions was observed for RPE ($p = 0.964$) or DOMS ($p = 0.732$), throughout the training intervention (Figs 12 & 13). The average RPE reported (Fig 13), across the seven-week training period were 5.75±1.26, 5.68±0.92 and 5.68±1.37, for the NHE, sprint and control training groups, respectively.

The average DOMS reported (Fig 14), across the seven-week training period were 3.16 ±1.36, 3.49±1.31 and 3.33±1.53, for the NHE, sprint and control training groups, respectively.

## Discussion

The results of the present study demonstrate that a multi-modal approach to hamstring training is highly effective in increasing both the modifiable risk factors of HSI (eccentric hamstring strength and BF$_{LH}$ FL), while being included within an ecologically valid training intervention that aided in increasing athletic performance.

**Table 4. Pairwise comparisons of body mass for all training groups.**

| | Body Mass (kg) | | | | |
|---|---|---|---|---|---|
| Group | Pre | Post | Mean Difference (%) | Hedge's g (95% CI) | p |
| Nordic | 75.47 ± 11.39 | 77.06 ± 15.97 | 1.74 (2.30) | 0.09 (-2.49–2.83) | 0.734 |
| Sprint | 71.36 ± 9.11 | 74.13 ± 8.74 | 3.52 (4.93) | 0.34 (-0.15–1.05) | 0.227 |
| Control | 78.01 ± 81.58 | 81.58 ± 11.46 | 3.57 (4.58) | 0.29 (-0.11–0.93) | 0.179 |

**Table 5. Pairwise comparisons of Bicep femoris fascicle length for all training groups.**

| Absolute bicep femoris long head fascicle length (cm) | | | | |
|---|---|---|---|---|
| Group | Pre | Post | Mean Difference (%) | Hedge's *g* (95% CI) | *p* |
| Nordic | 9.85 ± 1.20 | 11.12 ± 0.88 | 1.26 (12.83) | 1.19 (0.87–1.54) | <0.001 |
| Sprint | 9.76 ± 0.74 | 10.71 ± 0.85 | 0.94 (9.67) | 1.16 (0.95–1.37) | <0.001 |
| Control | 9.66 ± 0.93 | 10.54 ± 0.94 | 0.88 (9.09) | 0.92(0.64–1.34) | <0.001 |
| Relative bicep femoris long head fascicle length | | | | |
| Group | PRE | POST | Mean Difference (%) | Hedge's *g* (95% CI) | *p* |
| Nordic | 0.22 ± 0.02 | 0.25 ± 0.02 | 0.03 (12.78) | 1.22 (0.82–1.56) | <0.001 |
| Sprint | 0.24 ± 0.02 | 0.26 ± 0.02 | 0.02 (9.24) | 1.09 (0.75–1.44) | <0.001 |
| Control | 0.21 ± 0.03 | 0.23 ± 0.03 | 0.02 (9.23) | 0.60 (0.35–0.98) | <0.001 |

## Modifiable risk factors

The results of the present study identified meaningful increases (i.e., >SDD) for the modifiable risk factors (absolute and relative $BF_{LH}$ FL and eccentric hamstring strength) for all training groups, unsurprisingly the smallest magnitude of increase was observed within the control group–which for the present study was our single modality intervention, demonstrating the additional benefits that can be achieved with a multi-modal approach. Across the literature, the present study is the only study to date that has included HSI prevention, such as the NHE and sprinting, within a complete standardised training programme including a hip-dominant modality (RDL) which adds novelty to the literature.

The observed changes seen within the present study for $BF_{LF}$ FL are consistent with some of previous literature, there are some notable differences. A lot of the decisions made with regards to the intervention study design including; training volume, progressions, duration, and exercise selection (sprint and hip dominant exercise), were made from an Australian research group [1, 11, 13, 49], across these studies the magnitude in changes observed were greater than those within the present study, except for body weight alone NHE prescription [49]. However, the absolute changes in FL observed within the present study for the NHE group and those performed by the Australian research group, 1.26 cm vs 1.40–2.22 cm [1, 11, 13, 49], could be considered similar particularly given the differences in intensity and volume. Furthermore, the associated error within the measurement and estimation of $BF_{LH}$ FL that these studies employed could have influence the magnitude-based approach [54, 55]. Within the present study the associated error of FL measurement and estimation was mitigated by

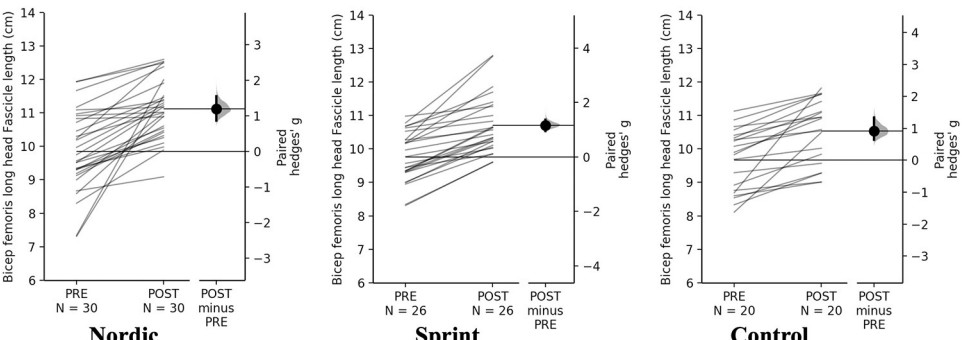

**Fig 2. Gardner-Altman estimation plots identifying Pre- and Post-intervention individual changes for absolute bicep femoris fascicle length and Hedge's *g* effect size with the 95% CI indicated by the ends of the vertical error bar.**

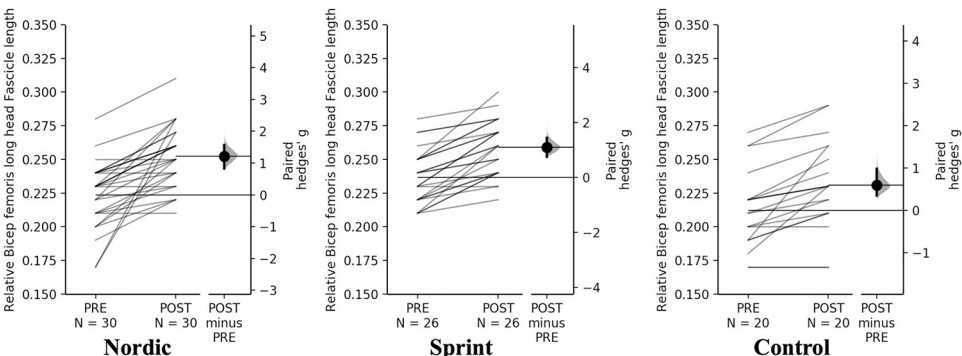

**Fig 3. Gardner-Altman estimation plots identifying Pre- and Post-intervention individual changes for relative bicep femoris fascicle length and Hedge's *g* effect size with the 95% CI indicated by the ends of the vertical error bar.**

utilising a 10 cm FOV. Contrastingly, Mendiguchia, Conceicão [24] found a mean difference of 1.66 cm from a sprint training intervention, which was considerably larger than what was found within the present study for the sprint group, 0.94 cm. Despite some methodological similarities, intervention duration and frequency, the exact prescription was vastly different, including both greater volumes of both sprint assistance work (i.e., resisted sprint work and plyometrics) and greater volumes of maximal effort sprints. Finally, the control group performing the RDL had a moderate increase in BF$_{LH}$ FL, recent literature only observed a small effect when using a stiff leg deadlift [56]. However, again a shorter intervention duration and the associated error of FL measurement and estimation could explain the difference in observed adaptations.

Consistent with previous training interventions, absolute and relative eccentric hamstring strength was increased across all training groups [1, 23, 28, 32, 49]. The eccentric hamstring strength changes observed for the NHE training group were larger than those highlighted within previous literature [1, 23, 28, 32, 49], however there are potential explanations as to why these studies may have observed small adaptations. Firstly, Pollard, Opar [49] and Suarez-Arrones, Lara-Lopez [32] used strong and extremely strong participants; with initial eccentric hamstring scores of 440–460 N and 570-692N for Pollard, Opar [49] and Suarez-Arrones, Lara-Lopez [32], respectively. This indicates that the magnitude of any adaptations for the stronger athletes would be smaller across any intervention [57]. Across the remaining literature where the present study presented greater adaptations [1, 23, 28], there is the potential for methodological dissimilarities having a pronounced effect. Specifically, both Freeman, Young

**Table 6. Pairwise comparisons of eccentric hamstring measures for all training groups.**

| Absolute peak eccentric hamstring strength (N) | | | | |
|---|---|---|---|---|
| **Group** | **Pre** | **Post** | **Mean Difference (%)** | **Hedge's *g* (95% CI)** | ***p*** |
| **Nordic** | 317.71 ± 61.93 | 431.28 ± 59.86 | 113.58 (35.75) | 1.84 (1.31–2.55) | <0.001 |
| **Sprint** | 295.80 ± 72.90 | 386.00 ± 54.51 | 90.20 (30.49) | 1.38 (1.06–1.78) | <0.001 |
| **Control** | 312.50 ± 70.98 | 351.91 ± 57.47 | 39.41 (12.61) | 0.60 (0.31–0.87) | 0.001 |
| Relative peak eccentric hamstring strength (N/kg) | | | | |
| **Group** | **PRE** | **POST** | **Mean Difference (%)** | **Hedge's *g* (95% CI)** | ***p*** |
| **Nordic** | 4.27 ± 0.83 | 5.69 ± 0.79 | 1.42 (33.15) | 1.72 (1.21–2.40) | <0.001 |
| **Sprint** | 3.35 ± 0.83 | 4.48 ± 0.63 | 1.12 (33.44) | 1.50 (1.17–1.91) | <0.001 |
| **Control** | 3.14 ± 0.71 | 3.56 ± 0.58 | 0.42 (13.44) | 0.63 (0.35–0.91) | 0.001 |

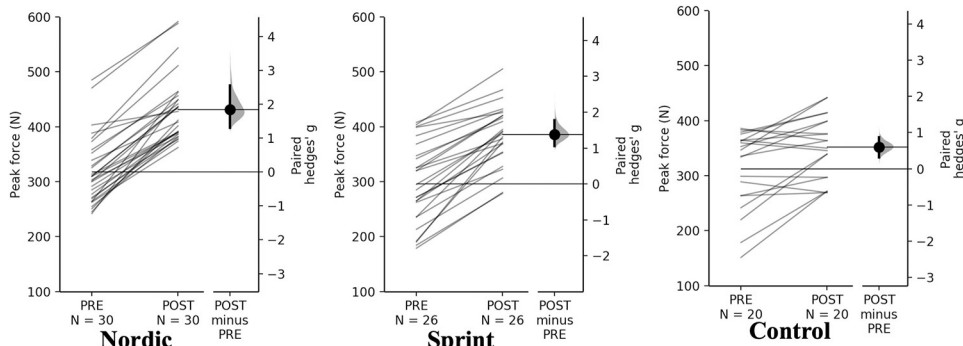

**Fig 4. Gardner-Altman estimation plots identifying Pre- and Post-intervention individual changes for peak eccentric hamstring force and Hedge's *g* effect size with the 95% CI indicated by the ends of the vertical error bar.**

[23] and Ishoi, Holmich [28] had no progression of intensity, which is a key factor in achieving eccentric adaptation. Furthermore, despite Bourne, Duhig [1] progressing the eccentric intensity with the addition of load–the prescription could have been excessive with high volumes. This is highlighted by Presland, Timmins [13] and Cadu, Goreau [15], who have both used low and extremely low session volumes and observed increased in eccentric hamstring strength. Although the literature is contradictory on these low volumes of NHE [14]. However, it should be noted that Presland, Timmins [13] and Siddle, Weaver [14] implemented a high volume initial standardised programme, which could have resulted in supercompensation, although this was only maybe present in the earlier study by Presland, Timmins [13]. Applying this initial standardised programme would be impossible within practice, with the additional effect of DOMS and the potential interference with sport-based training.

With regards to other modalities used within this study (i.e. sprint and hip dominant traditional exercise), the present study found a greater change in eccentric hamstring strength than Freeman, Young [23]. The present study utilised a standardised multi-modal prescription, whereas further training was not standardised by Freeman, Young [23], furthermore, the study by Freeman, Young [23] was of a short duration both of these factors could have influenced the observed adaptations. In contrast, the control group who only performed the RDL as part of their standardised training, had small increases in eccentric hamstring strength (39.41 N). This is consist with recent literature using the stiff leg deadlift, where a small increase in isokinetic eccentric hamstring strength was observed [56], however, in contrast,

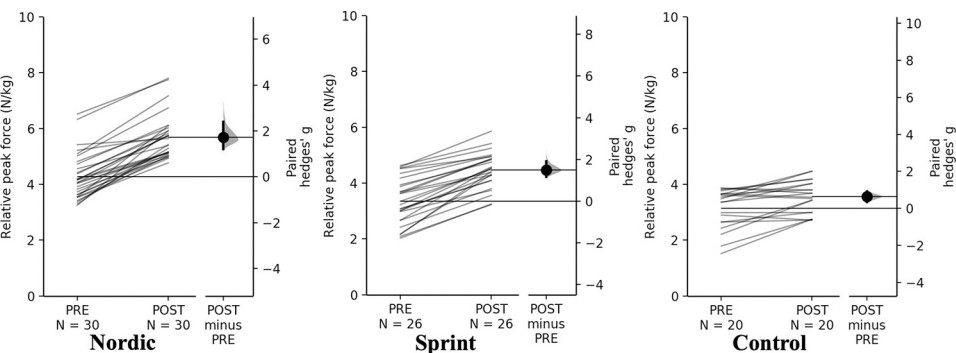

**Fig 5. Gardner-Altman estimation plots identifying Pre- and Post-intervention individual changes for relative peak eccentric hamstring force and Hedge's *g* effect size with the 95% CI indicated by the ends of the vertical error bar.**

Table 7. Pairwise comparisons of countermovement jump measures for all training groups.

| Take-off velocity (m·s⁻¹) | | | | | |
|---|---|---|---|---|---|
| **Group** | **PRE** | **POST** | **Mean Difference (%)** | **Hedge's g (95% CI)** | **P** |
| **Nordic** | 2.56 ± 0.24 | 2.67 ± 0.20 | 0.11 (4.44) | 0.48 (0.27–0.74) | <0.001 |
| **Sprint** | 2.43 ± 0.20 | 2.54 ± 0.16 | 0.11 (4.57) | 0.64 (0.24–1.28) | <0.001 |
| **Control** | 2.46 ± 0.25 | 2.59 ± 0.26 | 0.13 (5.15) | 0.48 (0.34–0.63) | 0.001 |
| **Jump momentum (kg· m·s-1)** | | | | | |
| **Group** | **PRE** | **POST** | **Mean Difference (%)** | **Hedge's g (95% CI)** | **P** |
| **Nordic** | 193.51 ± 33.33 | 206.02 ± 48.65 | 14.34 (7.41) | 0.29 (0.03–0.78) | 0.154 |
| **Sprint** | 172.44 ± 31.80 | 188.32 ± 28.59 | 14.97 (8.64) | 0.57 (0.08–1.29) | 0.045 |
| **Control** | 195.07 ± 41.37 | 213.71 ± 38.21 | 19.11 (9.96) | 0.45 (0.13–0.87) | 0.013 |

Bourne, Duhig [1] using the 45˚ hip extension found a large increase in eccentric hamstring strength (110.47 N). Although Bourne, Duhig [1] utilised greater training volumes and intervention duration potentially explaining the difference. Furthermore, the present study capped intensity at ~75% 1RM, whereas to aid in strength development a greater relative intensity could have been prescribed, more in line with strength training recommendations [12].

## Athletic performance

Meaningful increases in CMJ take-off velocity were observed for all training groups. The increase in take-off velocity, would also represent an increased jump height, although the smaller measurement error observed with take-off velocity means the increases observed are less likely to be an effect of random error. It should be noted however, that the control group had the largest increase in CMJ take-off velocity, although the magnitude of increases was similar between groups. The addition of sprinting or NHE had less of an effect on jumping than the control training programme, suggesting the benefits to performance came from the conventional resistance program including the RDL. Non-significant changes were observed within mean propulsion force for the sprint and control training groups; however, all three training groups had a small increase in mean propulsion force to a similar magnitude, with the sprint training group having the greatest magnitude of adaptation. However, on an individual basis within the NHE training group, all bar one individual, which was within SEM, had a positive and meaningful increase within mean propulsion force. Whereas for both the sprint and control the individual response was mixed. This indicates that the NHE potentially led to an

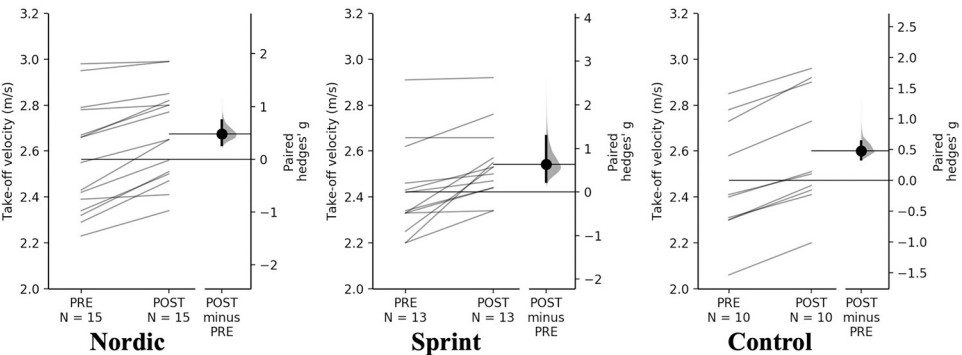

**Fig 6. Gardner-Altman estimation plots identifying Pre- and Post-intervention individual changes for take-off velocity and Hedge's g effect size with the 95% CI indicated by the ends of the vertical error bar.**

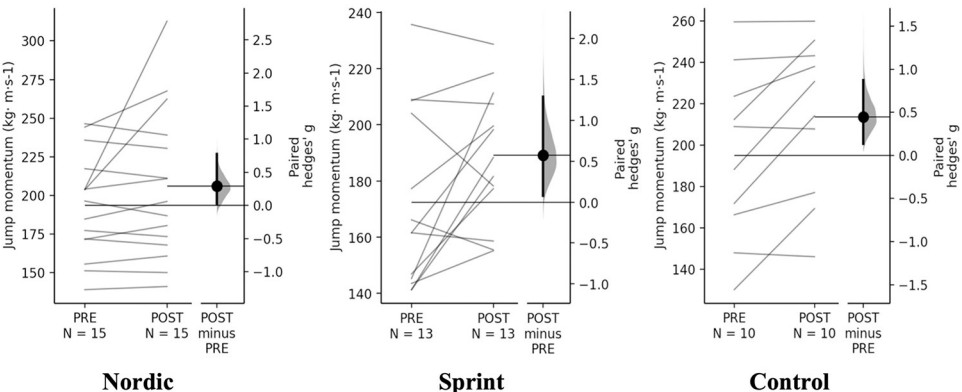

**Fig 7. Gardner-Altman estimation plots identifying Pre- and Post-intervention individual changes for jump momentum and Hedge's g effect size with the 95% CI indicated by the ends of the vertical error bar.**

increased force generating capacity during hip extension [1, 33]. However, for all other CMJ measures assessed, there were non-significant trivial differences from PRE to POST, this contradicts the hypothesis with no changes in the countermovement or braking phases, therefore adaptations to the hamstring architecture and eccentric hamstring strength had no influence on the ability to rapidly resist the downward motion during knee and hip flexion [58].

The control group had non-meaningful (<SDD) increase in absolute and relative peak net force attained during the IMTP, with a trivial and non-significant increase for absolute peak net force and small, significant increase in relative peak net force. Both NHE and sprint intervention groups, had meaningful (>SDD), significant and small increases in both absolute and relative peak net force. The sprint training group had the largest positive increase in both absolute and relative peak net force, 34.71- and 35.73%, respectively. Followed by the NHE training group had large positive increases in both absolute and relative peak net force, 22.28- and 22.46%, respectively. The observed increases in the sprint training group could be the result of increased potential of increase motor unit activation, increase passive tension of the muscle-tendon complex and improved cross bridge mechanics [22]. As was observed with the CMJ for NHE group, there may be an increased force generating capacity during hip extension as a result of the NHE exercise [1, 33], despite the IMTP is primarily a vertical, knee extension based task–the multi-joint nature of the task could explain the increases for the NHE group. The non-meaningful increase within the control group was surprising as all three groups

**Table 8. Pairwise comparisons of peak net IMTP force for all training groups.**

| Peak absolute net force (N) | | | | | |
|---|---|---|---|---|---|
| Group | Pre | Post | Mean Difference (%) | Hedge's *g* (95% CI) | *p* |
| Nordic | 1479.28 ± 804.67 | 1838.05 ± 603.80 | 329.64 (22.28) | 0.44 (0.28–0.65) | **<0.001**\* |
| Sprint | 1206.78 ± 743.58 | 1625.74 ± 775.07 | 418.95 (34.71) | 0.47 (0.33–0.68) | **<0.001**\* |
| Control | 1999.18 ± 482.45 | 2140.52 ± 472.64 | 141.35 (7.07) | 0.22 (0.12–0.37) | 0.619 |
| Peak relative net force (N/Kg) | | | | | |
| Group | Pre | Post | Mean Difference (%) | Hedge's *g* (95% CI) | *p* |
| Nordic | 18.62 ± 9.24 | 23.39 ± 5.72 | 4.18 (22.46) | 0.60 (0.29–0.98) | **<0.001**\* |
| Sprint | 16.72 ± 9.61 | 22.7 ± 9.12 | 5.98 (35.73) | 0.58 (0.36–0.88) | **<0.001**\* |
| Control | 26.06 ± 4.34 | 28.34 ± 4.07 | 2.28 (8.76) | 0.48 (0.25–0.80) | **0.034**\* |

\* = significant increase

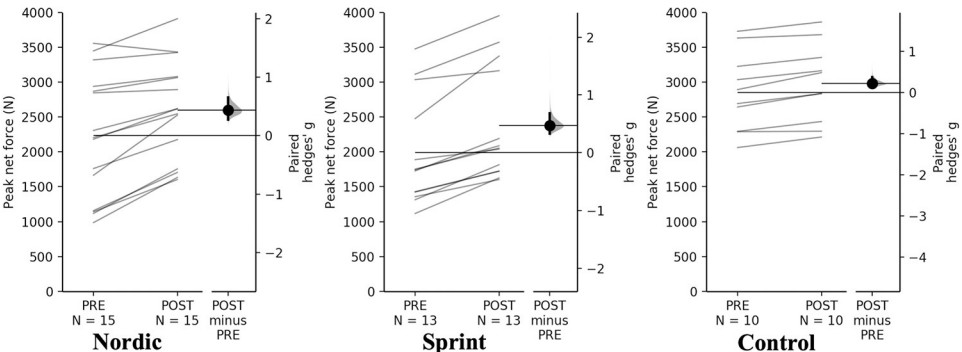

**Fig 8. Gardner-Altman estimation plots identifying Pre- and Post-intervention individual changes for peak net force and Hedge's *g* effect size with the 95% CI indicated by the ends of the vertical error bar.**

followed the same resistance training programme, therefore it was hypothesised that the same magnitude of increase would be seen for all training groups for the net force attained using the IMTP. Although, it should be noted that the control started and finished stronger than both the NHE and sprint training groups, therefore it could be expected that the magnitude of adaptations would be smaller when using any intervention for the control group [57].

The NHE and sprint training groups had meaningful and significant decreases in 0–10 m, 0–20 m, and 10–20 m sprint times. Across all sprint times, the sprint training group achieved the greatest decreases in comparison to the NHE training group. Although the differences cannot be entirely attributed to the NHE or sprint training, due to the accompanying resistance training programme [22, 26, 27]. Across all distances both groups had a near identical mean decrease (Table 9), this change in sprint ability from both groups could be an effect of two different mechanisms including; greater force generating capacity during hip extension as a result of the NHE exercise [1], which is specific to acceleration based tasks [33]. The sprint training group could have had improved structural and functioning properties of the muscle which could account for improvements such as, strengthened elastic properties of connective tissue; increase motor unit activation; increase passive tension of the muscle-tendon complex and improved cross bridge mechanics [22, 26, 27]. The decrease in sprint times for the NHE group is similar to what has been reported previously in a systematic review and meta-analyses, where a statistically significant decrease of -0.04 sec across all distances (5 m, 10 m and 20 m) in all studies included [59]. With even small improvements as those observed being greater

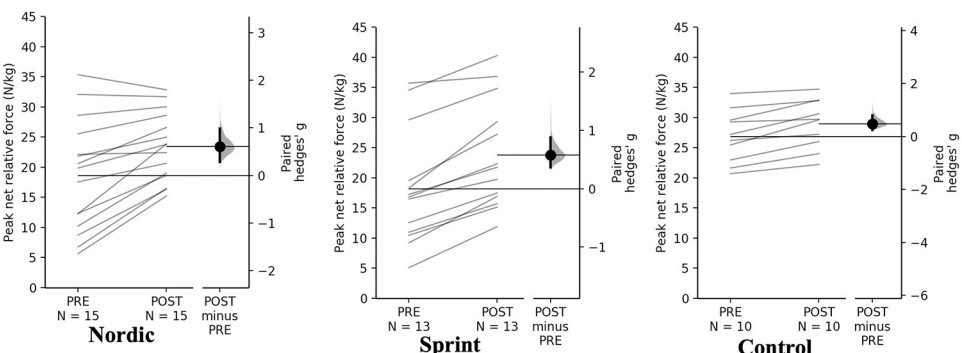

**Fig 9. Gardner-Altman estimation plots identifying Pre- and Post-intervention individual changes for peak net relative force and Hedge's *g* effect size with the 95% CI indicated by the ends of the vertical error bar.**

**Table 9. Pairwise comparisons of sprint measures between Nordic and sprint training groups.**

| | 0-10m time (s) | | | | |
|---|---|---|---|---|---|
| Group | Pre | Post | Mean Difference (%) | Hedge's *g* (95% CI) | *p* |
| Nordic | 1.98 ± 0.13 | 1.90 ± 0.11 | -0.08 (-4.04) | -0.69 (-1.20 to -0.33) | 0.001 |
| Sprint | 1.96 ± 0.11 | 1.88 ± 0.08 | -0.08 (-4.08) | -0.76 (-1.29–0.42) | 0.002 |
| | 0-20m time (s) | | | | |
| Nordic | 3.35 ± 0.19 | 3.22 ± 0.17 | -0.13 (-3.88) | -0.67 (-0.92 to -0.38) | <0.001 |
| Sprint | 3.34 ± 0.27 | 3.20 ± 0.20 | -0.14 (-4.19) | -0.68 (-1.14 to -0.36) | <0.001 |
| | 10-20m time (s) | | | | |
| Nordic | 1.35 ± 0.08 | 1.31 ± 0.09 | -0.04 (-2.96) | -0.47 (-0.83 to -0.26) | 0.010 |
| Sprint | 1.38 ± 0.17 | 1.31 ± 0.12 | -0.07 (-5.07) | -0.71 (-0.98 to -0.45) | <0.001 |

than the SDD, suggesting they are practically relevant and potentially decisive in one on one duels in sporting situations [59]

## Compliance

The present study was highly effective at increasing both modifiable risk factors of HSI (eccentric hamstring strength of $BF_{LH}$ FL), as well as increasing athletic performance. One potential explanation as to why this study had such positive effect was that it achieved 100% compliance. The low volume approach utilised within the present study also limited the effect of DOMS with only moderate DOMS and RPE reported (Figs 13 and 14), even as participants were progressed up to higher eccentric intensities. Notably, the individual DOMS ratings did not decrease during the intervention period, contrasting much of the literature regarding repeated bout effect [60, 61]. This observation could be due to the DOMS rating being of a total body soreness rather than specifically to the hamstrings, which was thought to be more relevant to practice and sport. A minimum of 75% compliance was demonstrated to have most positive beneficial effect of HSI incidence [7], and it would be suspected that a similar finding would be observed for the modifiable risk factors of HSI [6]. With regards to application, a low volume approach to the NHE and sprinting used within

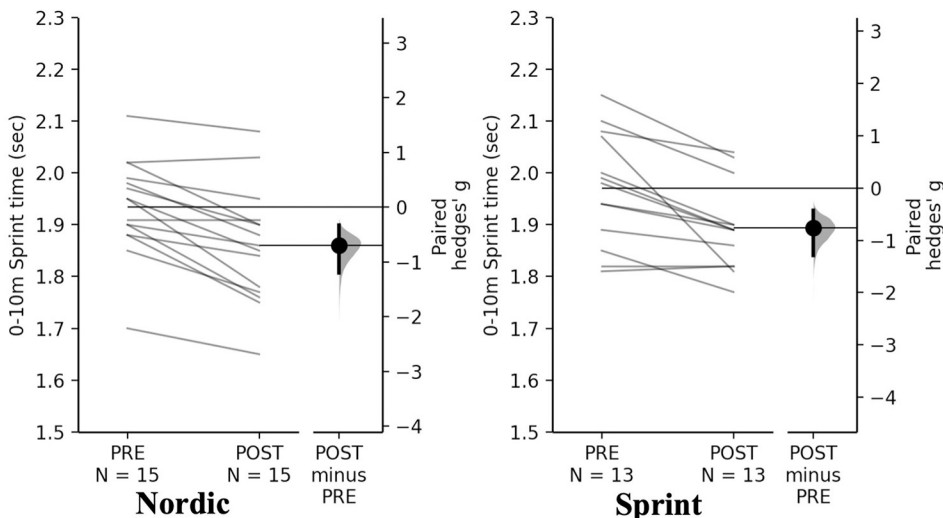

**Fig 10. Gardner-Altman estimation plots identifying Pre- and Post-intervention individual changes for 0-10m sprint time and Hedge's *g* effect size with the 95% CI indicated by the ends of the vertical error bar.**

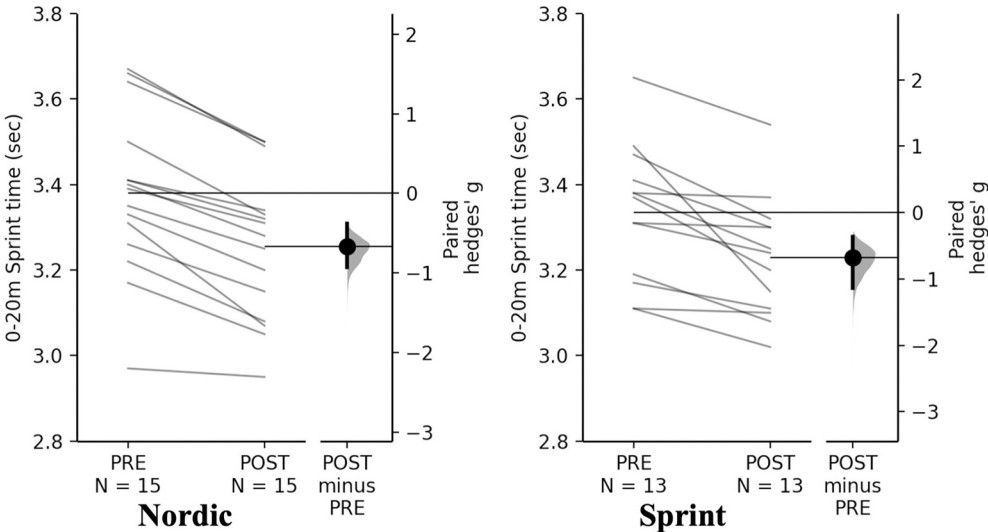

**Fig 11. Gardner-Altman estimation plots identifying Pre- and Post-intervention individual changes for 0-20m sprint time and Hedge's *g* effect size with the 95% CI indicated by the ends of the vertical error bar.**

practice could achieve greater volumes of compliance, specifically as a low volume NHE appears to have minimal influence DOMS. Additionally, sprinting can be made competitive, with immediate feedback further enhancing the positive experience that athletes can have when performing sprint training, increasing athlete compliance, with high levels of compliance (>80%) observed for sprint training [24]. However, a similar intervention using bounding could not achieve high levels of compliance, failing to reach the 75% identified, achieving a moderate level of compliance of 71% where a bounding exercise programme did not prevent HSI incidence [62].

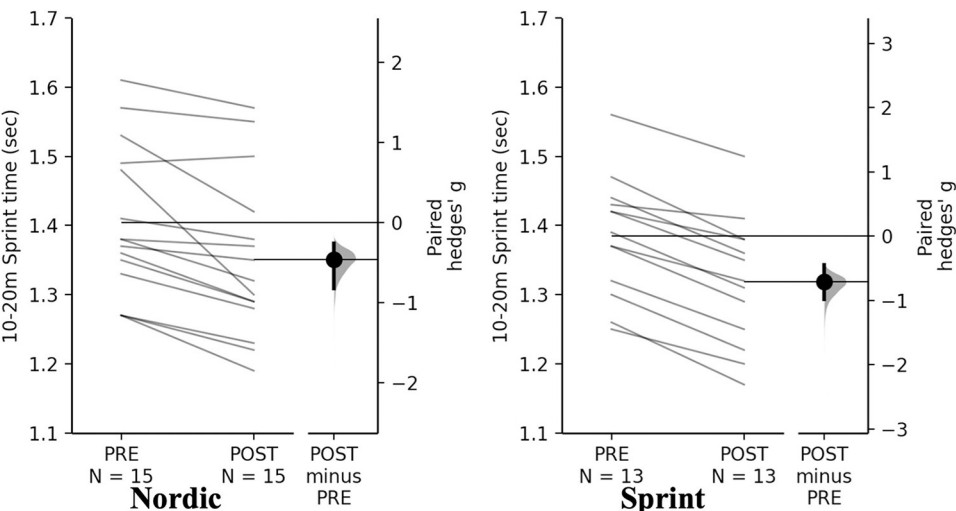

**Fig 12. Gardner-Altman estimation plots identifying Pre- and Post-intervention individual changes for 10-20m sprint time and Hedge's *g* effect size with the 95% CI indicated by the ends of the vertical error bar.**

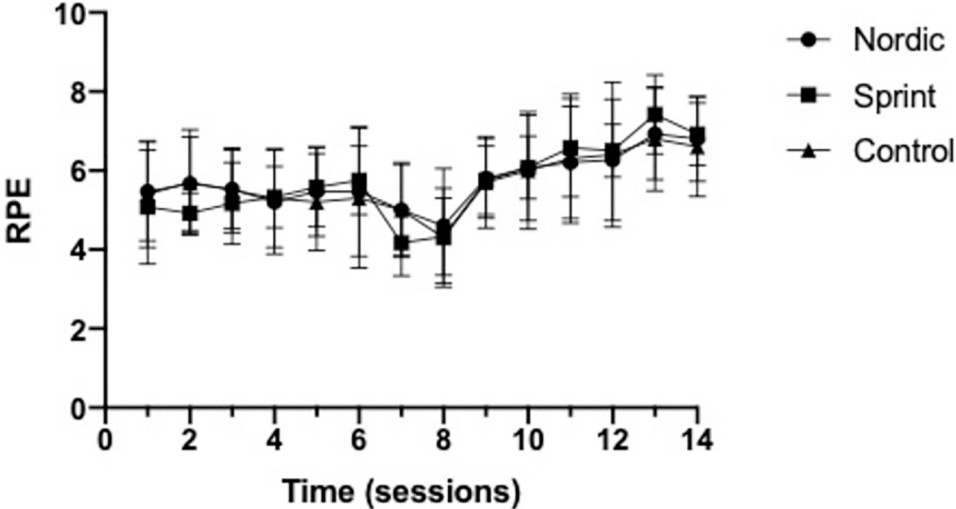

**Fig 13. Mean (±95%CI) Rating of perceived exertion measured using a numeric scale (1–10) for the Nordic hamstring exercise, Sprint and control groups.**

## Limitations

The present study is not without its limitations; firstly, although all participants reported participation in regular sport (predominantly team sport); competitive level, season, positional demands could have influenced the adaptations. This meant that individuals would have been exposed to a variety of external running and training loads, which could have all influenced the individual responses observed during the intervention [23, 63]. Despite the non-standardised nature of external training, both eccentric hamstring strength and $BF_{LH}$ FL saw increases across the sample. Although, this also highlights a strength of the present study as it

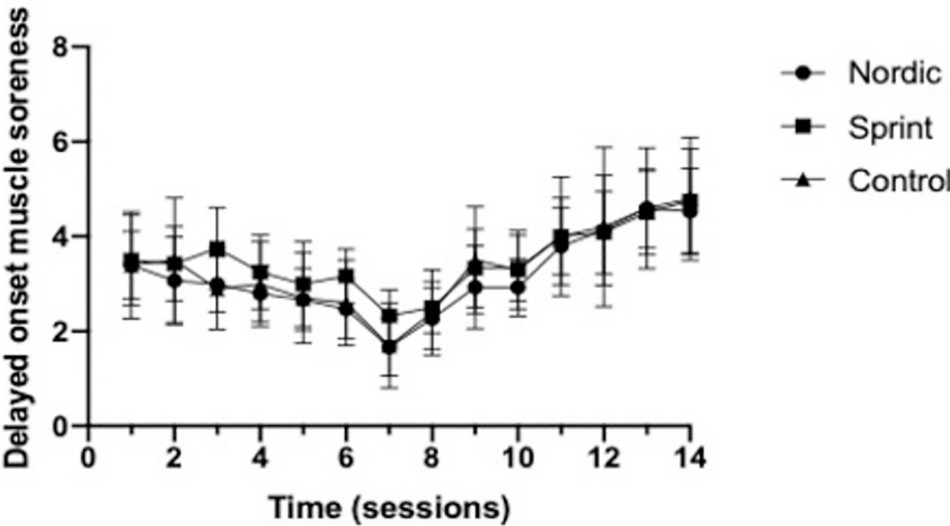

**Fig 14. Mean (±95%CI) 24-hr post soreness measured using a numeric pain rating scale (1–10) for the Nordic hamstring exercise, Sprint, and control groups.**

ecologically valid, as it based within a complete resistance training programme, where individuals were still participating within sport. A further limitation of the present study was that the assessment of eccentric hamstring strength was made using the NHE, i.e., training for the test rather than the potential adaptation. Especially as there is limited agreement between the Nordbord and isokinetic methods of hamstring assessment [64]. It is suggested future research should look to assess using a variety of other measures of hamstring strength such as isometric and isokinetic eccentric hamstring strength, to have a more comprehensive understanding of the eccentric adaptations to the training program.

The application of the training intervention could have been improved with appropriate feedback or technical modification. Real time visual feedback has been previously shown to increase mean eccentric peak force in the NHE within athletes [65], with suggestions that this could improve the adaptive response. Therefore, over the extended period used within the present study, the use of augmented real time feedback could have resulted in even greater adaptations than those presently observed. Additionally, the sprint training groups' application could have been improved by the utilisation of various drills and video feedback which could aid in technical modification. Although some of this may have added to overall training volume (i.e., distance), it could enhance the technical proficiency of participants potentially having a greater positive effect upon observed adaptations, this lack of technical instruction could also explain why the improvements in sprint performance were similar between the experimental groups. This should look to be employed by researchers in future investigations, to maximise the potential adaptive response from sprint training. As well as attempts to identify optimal sprinting volumes, frequencies and modalities which all have a positive effect on increasing $BF_{LH}$ FL and eccentric hamstring strength.

Finally, due to track unavailability, the control group was not able to perform any sprint assessments, this means that the conclusions made about the effect of sprint and NHE training upon improvement in sprint ability should be taken with caution. As the effect of the standardised training programme were not identified, as it would be expected increases in strength (i.e., IMTP peak net force), through the periodized resistance training programme could also transfer to sprint performance.

## Conclusions

The present study set out to determine the effect of a short-term training intervention with supplemental sprint or NHE, imbedded within an ecologically valid lower limb training programme, on the magnitude of adaptations to the modifiable risk factors of HSI, $BF_{LH}$ muscle architecture and eccentric hamstring strength, and measures of athletic performance. The findings of the present study highlight that utilising the NHE in addition to the lower limb training programme results in the meaningful increases in $BF_{LH}$ FL and eccentric hamstring strength, to a greater magnitude to the sprinting and control groups. Further inspection demonstrated that on an individual level all participants from each group increased $BF_{LH}$ FL and eccentric hamstring strength, with sprinting also being superior to the control intervention. The present study is the first of its kind to the authors knowledge to identify that a multimodal approach to training has the greatest positive effect upon modifiable risk factors of HSI. This is a crucial practical application for strength and conditioning coaches, sports rehabilitators, and sport scientists, in that HSI prevention should not come in a single form–it should form part a multimodal prescription containing multiple elements (e.g., NHE, sprinting and hip dominant exercises) and that hamstring strain injury prevention should not be performed in isolation. However, as the NHE was still superiorly effective in comparison to sprinting and the lower limb training programme alone, it maybe preferential to incorporate the NHE to

reduce the risk of HSIs. Further investigation is however warranted on the application of other supramaximal eccentric modalities other than the NHE due to the low levels of compliance reported in sport [9, 66]. However, sprinting could offer a viable alternative to supramaximal eccentric modalities at time points of the season, as long as acute sprint training loads do not compromise athlete readiness, potentially increasing risk of injury [35]. Although further investigation is required on the use of a more global sprint training intervention, such as the implementation of technical drills and resisted or assisted sprint efforts. Finally, there were improvements in measures of athletic performance across all training groups, highlighting that injury risk reduction and improved athletic performance can be achieved concurrently with the right application of training.

## Supporting information

**S1 Data.**
(XLSX)

## Author Contributions

**Conceptualization:** Nicholas J. Ripley, Matthew Cuthbert, Paul Comfort, John J. McMahon.

**Data curation:** Matthew Cuthbert.

**Formal analysis:** Nicholas J. Ripley, Matthew Cuthbert, Paul Comfort, John J. McMahon.

**Funding acquisition:** Nicholas J. Ripley.

**Investigation:** Nicholas J. Ripley, Paul Comfort, John J. McMahon.

**Methodology:** Nicholas J. Ripley, Matthew Cuthbert, John J. McMahon.

**Project administration:** Nicholas J. Ripley, Paul Comfort.

**Resources:** Matthew Cuthbert.

**Supervision:** Paul Comfort, John J. McMahon.

**Writing – original draft:** Nicholas J. Ripley.

**Writing – review & editing:** Nicholas J. Ripley, Paul Comfort, John J. McMahon.

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
