## [Decision Letter · Decision Letter 0]

3 Jan 2023

PONE-D-22-33022Effect of additional Nordic hamstring exercise or sprint training on the modifiable risk factors of hamstring strain injuries and performance.PLOS ONE

Dear Dr. Ripley,

Thank you for submitting your manuscript to PLOS ONE. After careful consideration, we feel that it has merit but does not fully meet PLOS ONE’s publication criteria as it currently stands. Therefore, we invite you to submit a revised version of the manuscript that addresses the points raised during the review process.

The reviewers agrees that the article have merit. However, some modifications should be done, namely in methods aiming to improve the replicability.

We look forward to receiving your revised manuscript.

Kind regards,

Filipe Manuel Clemente, PhD

Academic Editor

PLOS ONE

Journal Requirements:

3. Please upload a new copy of Figure 1 as the detail is not clear. Please follow the link for more information: https://blogs.plos.org/plos/2019/06/looking-good-tips-for-creating-your-plos-figures-graphics/"" https://blogs.plos.org/plos/2019/06/looking-good-tips-for-creating-your-plos-figures-graphics/

Reviewers' comments:

Reviewer's Responses to Questions

**Comments to the Author**

1. Is the manuscript technically sound, and do the data support the conclusions?

Reviewer #1: Yes

Reviewer #2: Yes

2. Has the statistical analysis been performed appropriately and rigorously? 

Reviewer #1: Yes

Reviewer #2: Yes

3. Have the authors made all data underlying the findings in their manuscript fully available?

Reviewer #1: Yes

Reviewer #2: No

4. Is the manuscript presented in an intelligible fashion and written in standard English?

Reviewer #1: Yes

Reviewer #2: Yes

5. Review Comments to the Author

Reviewer #1: Dear editor and authors,

The manuscript is at an advanced stage. It has a very well worked background and the experimental framework is very complete. The statistical analyzes are correct and very advanced and, in general, the manuscript needs few improvements to be published in the journal. The main errors found are:

1. There is a reference published in 2021 that should be used in this manuscript.

2. The problem of sample size has been dealt with correctly but in the wrong place. It has been recommended to the authors that they move it in the corresponding place.

3. The figures are pixelated and cannot be sent in this way for review. The authors must look for a solution.

4. The phrase of practical application should be further developed since it is very general and could be used in many manuscripts on this subject. Some issue that is an added value of this study should be incorporated.

Reviewer #2: This study aimed to to determine the effect of a lower-limb exercise program with

either additional Nordic Harmstring Exercise (NHE) or sprinting on modifiable risk factors of Harmstring Strain injury (HSI) and athletic performance. I consider that it is a study that deals with a hot topic in the literature. In addition, studies with intervention are an added value and well designed provide great application to training. Overall, the document is well referenced and very well written. I would like to congratulate the authors. Also, I have some doubts with results section. Likewise, I have some comments in order to improve the manuscript:

One of my questions is related to the inclusion criteria to be a subject of the study. In this sense, I wonder what happens to those players who are injured during the intervention period for a few days or weeks. I believe that a player who is injured for five weeks is not the same as five days. Has this aspect been taken into account when selecting the participants?

The exercises used in the training program should be referenced. In other words, it is convenient that the selection of the exercises have a proven scientific support.

Authors indicate that “Each resistance training session consisted of three lower limb exercises, where the training volume remained constant across the training intervention, whilst intensity was manipulated (Table 1).” Could specify the significance of percentage? How do calculate the 100%?

With the statistical analysis carried out, how do the authors know that there are no differences between the groups in the pre-test? This is a relevant aspect to test since the pre-post differences may be influenced by the initial level of the participants.

From my point of view, and considering that the reliability and measurement error are not objectives of the study, I consider that this results section should be included in methods section after each test. I want to know the opinion of authors. Also, if body mass results have been interpreted statistically, this variable should be appear in the objective of the study.

Results: I wonder why the authors indicate that the percentages are higher or lower in one group than in another, if they have not compared whether these percentages are statistically significant with respect to each other:

- “The Nordic group had the greatest percentage increase in absolute and relative BFLH FL in comparison to sprint and control groups, with only a small percentage differences between the sprint and control groups.”

- “[…] interestingly, the control group had the greatest percentage and magnitude of increase across training groups.”

- “The percentage changes for each group was similar for all splits, although the sprint group had marginally greater percentage decreases in split times.”

Discussion – Modifiable risk factors. Some modifications need to be made throughout this section. From my point of view, the key is to determine the reasons that can lead to understanding that adding additional exercises does not improve eccentric strength or hamstring architecture.

Discussion – Athletic performance. Nordic to improve velocity? Why doesn't the group that makes additional sprint content improve the sprint more?

Other some minor proposals of change:

Line 221: Delete: “, without the addition of the NHE or sprints,”

Line 232: It is difficult to understand that all athletes from different team sport modalities belong to semi-professional level. Please, concrete these competitive-level.

Line 239: Unify “participants” or “subjects”. Also, along the manuscript.

Table 7. Error in title. Revise, please. Delete: “Error! No text of specified style in document.”

6. PLOS authors have the option to publish the peer review history of their article (what does this mean?). If published, this will include your full peer review and any attached files.

Reviewer #1: **Yes: **Alfonso Castillo-Rodríguez

Reviewer #2: No

---

## [Author Response · Author response to Decision Letter 0]

20 Jan 2023

Review Comments to the Author and relevant replies, rebuttals or notification of changes.

Reviewer #1: Dear editor and authors,

1. There is a reference published in 2021 that should be used in this manuscript.

The authors are not sure which study you are referring too, if you could provide a reference, it would be appreciated, and we can look to include it within the manuscript. 

2. The problem of sample size has been dealt with correctly but in the wrong place. It has been recommended to the authors that they move it in the corresponding place.

This has been amended. Moved to Page 10 L433-440.

3. The figures are pixelated and cannot be sent in this way for review. The authors must look for a solution.

This has been amended. Hopefully these are sufficient.

4. The phrase of practical application should be further developed since it is very general and could be used in many manuscripts on this subject. Some issue that is an added value of this study should be incorporated.

This has been addressed and hopefully the updated applications now provide more value. Page 22 L11-37

Reviewer #2: 

One of my questions is related to the inclusion criteria to be a subject of the study. In this sense, I wonder what happens to those players who are injured during the intervention period for a few days or weeks. I believe that a player who is injured for five weeks is not the same as five days. Has this aspect been taken into account when selecting the participants?

A more thorough inclusion criteria has been identified (page 5 L229-245), additionally, none of the participants sustained any injury during the training intervention which has been clearly stated (page 5 L244-245)

The exercises used in the training program should be referenced. In other words, it is convenient that the selection of the exercises have a proven scientific support.

A reference (Comfort P, Thomas C, Dos’Santos T, Suchomel TJ, Jones PA, McMahon JJ. Changes in Dynamic Strength Indec in Response to Strength Training. Sports. 2018;6(176):1-10. doi: 10.3390.) has been included (page 6 line 253-255), demonstrating the training programmes positive beneficial effects of athletic performance. 34. 

Authors indicate that “Each resistance training session consisted of three lower limb exercises, where the training volume remained constant across the training intervention, whilst intensity was manipulated (Table 1).” Could specify the significance of percentage? How do calculate the 100%?

The participants 1RMs were estimated by individuals previous training history with each of the exercises, and then the submaximal percentages were based on this estimated value. This has been highlighted in detail within the manuscript (page 6 line 253-259).

With the statistical analysis carried out, how do the authors know that there are no differences between the groups in the pre-test? This is a relevant aspect to test since the pre-post differences may be influenced by the initial level of the participants.

Pre-intervention statistics have now been ran and included within the manuscript (page 13 line 3-8).

From my point of view, and considering that the reliability and measurement error are not objectives of the study, I consider that this results section should be included in methods section after each test. I want to know the opinion of authors. 

Although not an objective of the study, it is crucial information to collected on the included sample to be able to determine the smallest error of the measurement and smallest detectable difference. Essentially this informs if the changes were meaningful changes. Although as methods had been used previously and previous values have been identified, the authors believe it is still important to collect population specific SEM and SDD values rather than using previously published values. In the authors opinion, it is not of huge importance where the reliability statistics are included, only that they are included.

Also, if body mass results have been interpreted statistically, this variable should be appear in the objective of the study.

Body mass was interpreted statistically, as this is an important factor in determining changes in jump and sprint performance. Although not an objective of the study to increase or decrease body mass, any changes in body mass could explain the changes in jump performance (e.g. jump performance could decrease or remain constant but jump momentum improve due to increased mass), ideal outcomes would be for mass to stay constant or increase with improved jump performance i.e., jump height (take-off velocity) and jump momentum), the same could also be applied for sprint performance.

Results: I wonder why the authors indicate that the percentages are higher or lower in one group than in another, if they have not compared whether these percentages are statistically significant with respect to each other:

- “The Nordic group had the greatest percentage increase in absolute and relative BFLH FL in comparison to sprint and control groups, with only a small percentage differences between the sprint and control groups.”

- “[…] interestingly, the control group had the greatest percentage and magnitude of increase across training groups.”

- “The percentage changes for each group was similar for all splits, although the sprint group had marginally greater percentage decreases in split times.”

The data has been compared statistically using the methods described in the statistical approach, please see the associated p values within each of the tables identified within the results. The authors believe the magnitude (percentage and effect size) of change are more important in this instance, as a significant effect could be trivial in magnitude and therefore not have practical significance. This also falls in lines with recommendations from Cummings and Rivara (2003), Reporting Statistical Information in Medical Journal Articles, Arch Pediatr Adolesc Med, 157, 321-324. 

Discussion – Modifiable risk factors. Some modifications need to be made throughout this section. From my point of view, the key is to determine the reasons that can lead to understanding that adding additional exercises does not improve eccentric strength or hamstring architecture.

The authors are not sure on what the reviewer wants changing from this point, the section highlights that adding either additional NHE or sprinting improves fascicle length and eccentric hamstring strength to a greater extent than not including them. But the section also highlights that you do see some changes through the inclusion of a hip dominant based exercise, but including additional exercises is more effective. Additionally, it would be philosophised that practitioners should include a well-rounded approach to hamstring training, so highlighting why adding additional exercises does not improve the modifiable risk factors would contradict this while also contradicting the results.

Discussion – Athletic performance. Nordic to improve velocity? Why doesn't the group that makes additional sprint content improve the sprint more?

The question of the nordic improving running velocity has been explored previously in a meta-analysis by Bautista et al., (2021), observing that short sprint performance increases with the inclusion of the NHE. Additionally, as highlighted within the manuscript the observed changes in sprint performance could have been improved by the resistance training alone. A further discussion of potentially why both groups had similar improvements in performance has been added, with the potential for increased force generating capabilities in acceleration (Morin et al., 2015), especially around the hip. Moreover, the fact that the sprint group only performed sprinting with no technical or skill driven drills as discussed within the limitations suggesting that this could be another reason the improvements were similar, although the magnitude of change was larger in the 10-20m split, likely as a cause of the additional sprint work. 

Other some minor proposals of change:

Line 221: Delete: “, without the addition of the NHE or sprints,”

This has been amended. 

Line 232: It is difficult to understand that all athletes from different team sport modalities belong to semi-professional level. Please, concrete these competitive-level.

This has been explained in more detail, as it was a range from collegiate to semi-professional, not all were collegiate and not all were collegiate. It was just a range (page 5 line 229-233).

Line 239: Unify “participants” or “subjects”. Also, along the manuscript.

This has been amended throughout the manuscript.

Table 7. Error in title. Revise, please. Delete: “Error! No text of specified style in document.”

This has been amended.

---

## [Decision Letter · Decision Letter 1]

5 Feb 2023

Effect of additional Nordic hamstring exercise or sprint training on the modifiable risk factors of hamstring strain injuries and performance.

PONE-D-22-33022R1

Dear Dr. Ripley,

We’re pleased to inform you that your manuscript has been judged scientifically suitable for publication and will be formally accepted for publication once it meets all outstanding technical requirements.

Kind regards,

Filipe Manuel Clemente, PhD

Academic Editor

PLOS ONE

Additional Editor Comments (optional):

Reviewers' comments:

Reviewer's Responses to Questions

**Comments to the Author**

1. If the authors have adequately addressed your comments raised in a previous round of review and you feel that this manuscript is now acceptable for publication, you may indicate that here to bypass the “Comments to the Author” section, enter your conflict of interest statement in the “Confidential to Editor” section, and submit your "Accept" recommendation.

Reviewer #1: All comments have been addressed

Reviewer #2: All comments have been addressed

2. Is the manuscript technically sound, and do the data support the conclusions?

Reviewer #1: Yes

Reviewer #2: Yes

3. Has the statistical analysis been performed appropriately and rigorously? 

Reviewer #1: Yes

Reviewer #2: Yes

4. Have the authors made all data underlying the findings in their manuscript fully available?

Reviewer #1: Yes

Reviewer #2: Yes

5. Is the manuscript presented in an intelligible fashion and written in standard English?

Reviewer #1: Yes

Reviewer #2: Yes

6. Review Comments to the Author

Reviewer #1: (No Response)

Reviewer #2: Now, I fell comfortable with the review done. I think the manuscript have improved in this new version.

7. PLOS authors have the option to publish the peer review history of their article (what does this mean?). If published, this will include your full peer review and any attached files.

Reviewer #1: **Yes: **Alfonso Castillo-Rodriguez

Reviewer #2: No

---

## [Editor Report · Acceptance letter]

15 Feb 2023

PONE-D-22-33022R1 

Effect of additional Nordic hamstring exercise or sprint training on the modifiable risk factors of hamstring strain injuries and performance. 

Dear Dr. Ripley:

I'm pleased to inform you that your manuscript has been deemed suitable for publication in PLOS ONE. Congratulations! Your manuscript is now with our production department. 

Kind regards, 

on behalf of

Dr. Filipe Manuel Clemente 

Academic Editor

PLOS ONE